# The Physicochemical Properties and Antioxidant Activities of the Hawthorn Pectin Extracted Using Ultra-High Pressure Assisted Acid Extraction (UPAAE)

**DOI:** 10.3390/foods13070983

**Published:** 2024-03-23

**Authors:** Dihu Lv, Jianying Chen, Chun Yang

**Affiliations:** 1College of Food Science and Engineering, Shanxi Agricultural University, Jinzhong 030801, China; lvdihu@163.com (D.L.); 15388540129@163.com (J.C.); 2Shanxi Institute for Functional Food, Shanxi Agricultural University, Taiyuan 030001, China

**Keywords:** hawthorn pectin (HP), ultra-high pressure assisted acid extraction (UPAAE), acid extraction (AE), water extraction (WE), physicochemical properties, antioxidant activities

## Abstract

This study aims to investigate the positive effects of ultra-high pressure assisted acid extraction (UPAAE) on both physicochemical properties and antioxidant activities of hawthorn pectin. The basic indicators, structure characterization, and antioxidant activities were measured, which could indicate the disadvantages and advantages among traditional water extraction (WE), acid extraction (AE), and UPAAE. The results show that the hawthorn pectin of UPAAE has a decrease in esterification degree, protein content, and total polyphenols, but has an increase in total galacturonic acid aldehyde compared to the hawthorn pectin of AE. In the Fourier Transform Infrared Spectroscopy (FT-IR) and Scanning Electron Microscopy (SEM) analyses, the hawthorn of UPAAE has typical pectin absorption peaks in the FT-IR spectrum and a distinct layered structure in the SEM surface image. The ion chromatography profiles show that the molar ratio of galacturonic acid to arabinose in the hawthorn pectin of UPAAE increases and 5.50 μg/mg ribose appears compared to the pectin of AE and WE. The high performance gel permeation chromatography (HPGPC) profile indicates that the molecular weight distribution of hawthorn pectin of UPAAE is more concentrated and has the highest molecular weight compared to the pectin of the other two extraction methods. In the vitro antioxidant activity analysis, the pectin of UPAAE exhibits the highest scavenging rate against 1,1-diphenyl-2-picrylhydrazyl (DPPH) radicals (93.70%), which is close to the scavenging rate of vitamin C (96.30%). These findings demonstrated that UPAAE is a more efficient and environmentally friendly method for pectin extraction from hawthorn. It is also an effective way to enhance its antioxidant activity, which has great application prospects in the food industries.

## 1. Introduction

Hawthorn, a widely cultivated plant of the Crataegus genus, is considered as an alkaline food and has significant production in China. Throughout history, hawthorn has served as both a medicinal herb, aiding digestion and stimulating appetite, and as a fruit enjoyed prior to or following meals, earning widespread popularity. The hawthorn fruit is rich in various nutrients, including sugars, vitamins, dietary fiber, etc. [1]. Chen et al. mentioned that hawthorn is abundant in pectin content, reaching up to 9.94% [2], making it an ideal raw material for pectin extraction.

Pectin, as an acidic polysaccharide, exists in three forms in fruit tissues: protopectin, pectin, and pectic acid. It is one of the main components of higher plant cell walls. Due to its multifunctionality, natural origin, and environmentally friendly characteristics, pectin has found widespread applications in various industries. There is a significant global demand for pectin every year, and the majority of commercially available pectin is extracted from citrus peels and apple pomace. The production falls short of the demand, leading Chinese researchers in recent years to concentrate on developing pectin extraction from alternative sources such as dragon fruit peels [3], fig peels [4], grape pomace [5], and so on. Additionally, various extraction methods for pectin have been explored, including hot water extraction, ultrasound-assisted extraction, microwave-assisted extraction [4], and more. The water extraction (WE) method is the most traditional approach for pectin extraction. Although it has advantages such as simple operation, easy control of conditions, and low cost, the yield of pectin obtained using this method is low. The ultrasound-assisted method has the advantage of improving the extraction rate in a short time period [6], but this method easily leads to a decrease in polysaccharide activity, and if the ultrasonic time is too long, the raw material will be significantly broken, and other impurities will dissolve out, resulting in further purification difficulty [7]. The microwave-assisted method has the advantages such as shortening the process time, reducing energy consumption and improving the purity of polysaccharides [4]. However, if the wavelength and power are unsuitable, the differences between pectin products will be significant [8], the temperature will rise quickly, which is difficult to control, and the quality of pectin will be inconsistent [9]. Therefore, these methods have some limitations in the process of extracting pectin polysaccharides.

In 2001, ultra-high pressure (UHP) technology was approved by the U.S. Food and Drug Administration (FDA) for the processing of fruit and vegetable juices, followed by approval for use in ready-to-eat foods by the U.S. Department of Agriculture’s Food Safety and Inspection Service in 2004 [10]. In 2009, the FDA approved UHP technology for food sterilization. UHP technology not only provides effective sterilization but also minimally affects the physicochemical properties, sensory qualities, and nutritional composition of food, thereby preserving its original flavor to the greatest extent. In recent years, researchers have begun to use this technology for the extraction of natural products, such as polysaccharide extraction from *Porphyra haitanensis*, wheat germ and Cordyceps pupa [11,12,13], ginsenoside extraction from ginseng [14], phenolic substances from longan fruit pericarp [15], anthocyanins extraction from mulberry, etc. [16], but there has been no in-depth research on the extraction of hawthorn pectin. During the application of UHP technology, the significant pressure difference allows the solvent to rapidly penetrate the internal tissues of the raw materials. This facilitates the rapid dissolution of active ingredients in the extracting solvent, promoting thorough contact between the extract and the extracting solvent, thereby increasing the yield. The impact of ultra-high pressure technology on the extracted substances is generally less compared to other extraction methods [17]. Although there has been progress in the study of UHP equipment, it cannot be denied that the equipment itself is expensive, which determines that the current high cost of the technology makes it unsuitable for large-scale production in the food industry.

In this study, hawthorn powder is used as the raw material and ultra-high pressure assisted acid extraction (UPAAE) is employed to prepare hawthorn pectin. Simultaneously, this method is compared with traditional WE and AE in terms of physicochemical properties and antioxidant activity of the hawthorn pectin.

## 2. Materials and Methods

### 2.1. Materials and Chemicals

Fresh hawthorn (variety: Dajinxing) was procured in Nanma Village, Xiaodian District (Taiyuan, China) in early October 2022, with a moisture content of 79%. Citric acid, sodium hydroxide, salicylic acid, ferric chloride, and trichloroacetic acid were purchased from Damao Chemical Reagent Co., Ltd. (Tianjin, China). n-Butanol was obtained from Zhiyuan Chemical Reagent Co., Ltd. (Tianjin, China). Anhydrous ethanol was acquired from Yongda Chemical Reagent Co., Ltd. (Tianjin, China). Sodium tetraborate was sourced from Beijing Solaibao Biotechnology Co., Ltd. (Beijing, China). Chloroform and sulfuric acid were provided by the laboratory. Potassium ferrocyanide was procured from Tianjin Fengchuan Chemical Reagent Technology Co., Ltd. (Tianjin, China). Coomassie Brilliant Blue G250, bovine serum albumin, anthrone, gallic acid and ascorbic acid were purchased from Shanghai yuanye Bio-Technology Co., Ltd. (Shanghai, China). Except for gallic acid, which was used as a standard, all other chemical reagents were of analytical grade.

### 2.2. Preparation of Hawthorn Powder

Fresh hawthorn fruits with uniform shapes and sizes, free from diseases and pests were used. The surfaces of the fruits were cleaned, they were immersed in boiling water for 2 min (to deactivate the enzymes), quickly removed and allowed to cool, the seeds and stems were removed, the pulp collected, and they were cut into small pieces. The raw materials were then arranged evenly on an iron tray and placed in a freezer at −40 °C overnight for freezing. The samples were freeze-dried using a vacuum freeze dryer (SCIENTZ-10YD/A Freeze Dryer; Ningbo Xin Zhi Freeze Drying Equipment Co., Ltd., Ningbo, China) and the resulting powder was sieved through a 60-mesh screen. The powder was then sealed in bags and stored in the refrigerator for future use.

### 2.3. Traditional Water Extraction (WE)

The WE method is referenced from Hou et al. [18]. To extract the hawthorn powder, 20 g of freeze-dried powder was weighed in a conical flask and mixed with distilled water at a ratio of 1:15. The mixture was then subjected to hot water extraction at 90 °C for 4 h with continuous stirring to ensure thorough extraction. After extraction, the liquid was centrifuged at 5000 rpm for 20 min, and the resulting supernatant was concentrated to one-third of the original volume. Subsequently, the concentrated liquid underwent two rounds of protein removal using the Sevage method, followed by 72 h of dialysis (with a 3500 Da cutoff; Hunan Yibo Biotechnology Co., Ltd., Changsha, China). Afterward, alcohol precipitation was performed and the resulting precipitate was centrifuged at 5000 rpm for 10 min. The resulting pellet was dissolved in distilled water and then vacuum freeze-dried to obtain hawthorn pectin, named WE-HP.

### 2.4. Acid Extraction (AE)

The hawthorn pectin was extracted using the optimal conditions reported in the literature [19]. To prepare the hawthorn pectin named AE-HP, 10 g of freeze-dried hawthorn powder was weighed in a conical flask. Distilled water was added at a ratio of 1:30, adjusted to pH 2.5 with 1 mol/L citric acid. The mixture was then extracted in a 90 °C water bath for 150 min, following the traditional water extraction method.

### 2.5. Ultra-High Pressure Assisted Acid Extraction (UPAAE)

The hawthorn pectin was extracted using the optimal conditions for ultra-high pressure assisted acid extraction, as reported in the literature [19]. In total, 5 g of hawthorn powder was weighed in a conical flask and distilled water at a ratio of 1:62 with a pH of 2.5 (citric acid; 1 mol/L) was added. The extraction process was carried out in a 90 °C water bath for 150 min with continuous stirring. After extraction, the mixture was rinsed with tap water for 5 min. The solution was then poured into a pressure-resistant plastic bottle of appropriate size for high-pressure extraction at 450 MPa, with a holding time of 598 s (SHPP-8.8 L-600 MPa; Shanxi Sanshuihe Technology Co., Ltd., Taiyuan, China). The remaining steps followed the traditional water extraction method, resulting in the production of hawthorn pectin, named UPAAE-HP.

### 2.6. Basic Indicators of Hawthorn Pectin

#### 2.6.1. Degree of Pectin Esterification (DE)

Esterification degree is one of the important parameters for evaluating pectin. Different esterification degrees lead to variations in the physicochemical properties of pectin. The esterification degree of hawthorn pectin was determined using the titration method reported by Tuğba Öztürk [20] with slight modifications. The method is as follows: 10 mL of hawthorn pectin solution prepared using different extraction methods, add three drops of phenolphthalein solution (1%) as an indicator, and titrate with NaOH solution (0.001 mol/L) until the solution turns pink and remains unchanged for 30 s. Record the volume of NaOH solution consumed at this point as *V*_1_. Then add 2 mL of NaOH solution (0.005 mol/L) to the mixed solution, shake, and let it stand at room temperature for 15 min. Continue titrating with HCl solution (0.005 mol/L) until the pink color just disappears and remains unchanged for 30 s. Finally, add three drops of phenolphthalein indicator (1%) and titrate with NaOH solution (0.001 mol/L) until the solution turns pink and remains unchanged for 30 s. Record the volume of NaOH solution consumed at this point as *V*_2_.
(1)$$DE(\%)=\frac{V_{2}}{V_{1}+V_{2}}\times 100$$

#### 2.6.2. Total Galacturonic Acid Content (GalA)

The Gal A content in hawthorn pectin was determined using the carbazole-sulfuric acid method according to Chen [21]. In total, 0.5 mL of D-Gal A standard solution (20, 40, 60, 80, 100 mg/L, respectively) was placed in a stoppered test tube. Then, 3 mL of borax sulfuric acid solution (5 mg/mL) was added under ice-water bath conditions and mixed with a vortex mixer. The resulting mixture was kept in a boiling water bath for 5 min, then immediately cooled. After that, a 0.1 mL aliquot of the carbazole solution (0.1%) was added to the mixture, boiled for 5 min, cooled to room temperature, and the absorbance was measured at 530 nm. A total of 0.5 mL of sample solution was used instead of the standard solution and the absorbance value was measured.

#### 2.6.3. Protein Content

The protein content in hawthorn pectin was determined using the Coomassie Brilliant Blue method. A 10 mL stoppered test tube was filled with 1 mL of the protein standard solution (0, 20, 40, 60, 80, 100 mg/L, respectively). Then, 4 mL of Coomassie Brilliant Blue G-250 dye solution (0.01%) was added and mixed thoroughly. After 10 min at room temperature, the absorbance was measured at 595 nm. For the sample measurement, 1 mL of the sample solution was taken instead of the standard solution, using the blank reagent as a reference. Three replicates were performed to determine the protein content in the sample.

#### 2.6.4. Total Polyphenol Content (TPC)

The total polyphenols in hawthorn pectin were determined using the Folin–Ciocalteu method [22], with the following modifications to the procedure: 20 μL of gallic acid standard solution (40, 50, 80, 120, 140 mg/L) were mixed with 100 μL of Folin–Ciocalteu reagent (10%), and the mixture was stirred. After 5 min, 80 μL of Na_2_CO_3_ solution and 50 μL of water were added, and the solution was left at room temperature for 1 h. Then, 200 μL of the solution were transferred to a microplate and the absorbance was measured at 765 nm using a microplate reader (1510; Thermo Fisher Scientific Instrument Co., Ltd., Shanghai, China). In total, 20 μL of the sample solution were used instead of the standard solution and the absorbance was measured.

### 2.7. Structure Characteristics

#### 2.7.1. Fourier Transform-Infrared (FT-IR)

Hawthorn pectin freeze-dried samples extracted using different methods were mixed with KBr in a ratio of 1:100 and then ground uniformly. The resulting mixture was ground uniformly and spread evenly on a glass slide. A transparent pellet was then formed by pressing the mixture without causing any cracks. Next, the sample was placed in the holder of the FT-IR spectrometer (Nicolex is5; Thermo Fisher Scientific Instrument Co., Ltd., Shanghai, China) for scanning. The scanning range was set to 4000–400 cm^−1^, with a resolution of 4 cm^−1^ and 64 scans.

#### 2.7.2. Scanning Electron Microscopy (SEM)

Scanning electron microscopy is a commonly used technique for observing the microstructure and morphology of samples. It provides high magnification and a wide field of view, allowing for direct observation of the sample’s microstructure. In this study, freeze-dried samples of hawthorn pectin prepared using different extraction methods were placed on the sample stage and subjected to gold sputtering treatment. The microstructure of hawthorn pectin was observed using a scanning electron microscope (JSM-7500F; JEOL, Tokyo, Japan) at an acceleration voltage of 10 kV and magnifications of 100× and 5.0k×.

#### 2.7.3. Monosaccharide Composition

Monosaccharide composition was determined using the ion chromatograph method (ICS5000; Thermo Fisher Scientific) with 16 standard monosaccharides. In total, 0.5 mg of freeze-dried hawthorn pectin was hydrolyzed using 2 mL of 3 M TFA at 120 °C for 3 h in an ampoule. The acid hydrolysis solution was accurately pipetted into a tube, transferred to a nitrogen blower, and blow dried. Then, 5 mL of water was added and vortexed to mix, 50 μL pipetted, 950 μL of deionized water was added, and the solution was finally centrifuged. Then, ion chromatography analysis was performed. Equipped with a Dionex Carbopac^TM^ PA20 (3 × 150; Thermo Fisher Scientific, Waltham, MA, USA) chromatographic column and electrochemical detector, the mobile phase consisted of: H_2_O; 15 mmol/L NaOH; 10 mmol/L NaOH and 10 mmol/L NaAc. The analysis conditions are as follows: flow rate at 0.3 mL/min, injection volume of 5 μL, and column temperature at 30 °C.

#### 2.7.4. Molecular Weight Distribution

The molecular weight of pectin extracted through different methods was analyzed using high-performance liquid chromatography with a Shimadzu LC-10A system (Shimadzu, Kyoto, Japan). The system was equipped with a tandem gel column (BRT105-103-101, 8 mm × 300 mm) and a refractive index detector (RI-20A, Shimadzu).

A 0.05 M NaCl solution was used as the mobile phase. The hawthorn pectin freeze-dried sample was dissolved in the mobile phase to give a concentration of 5 mg/mL. After ultrasonic treatment for 10 min and centrifugation, the supernatant was filtered through a 0.22 μm aqueous microporous membrane and then placed in a sample vial for detection. The injection conditions were as follows: injection volume 25 μL, column temperature 40 °C, flow rate 0.8 mL/min, and elution time 60 min.

### 2.8. In vitro Antioxidant Activity Determination

For in vitro antioxidant activity experiments, solutions with different concentration gradients (0.25, 0.5, 1, 2, 4 mg/mL) were prepared using hawthorn pectin extracted using different methods, with Vc serving as a positive control.

#### 2.8.1. 2,2-Diphenyl-1-Picrylhydrazyl (DPPH) Radical Scavenging Activity Determination

DPPH radical scavenging activity was determined with improvements based on the method by Gao et al. [23]. A total of 4.9 mg of DPPH was weighed and dissolved in anhydrous ethanol in a 50 mL brown volumetric flask. The solution was then diluted to the desired volume. Next, 0.14 mL of DPPH-anhydrous ethanol solution was added to the reaction system, followed by the addition of 0.14 mL of pectin solution at different concentration gradients. The reaction mixture was thoroughly mixed and left in the dark at room temperature for 30 min. This process was repeated in triplicate, and the absorbance values were measured at 517 nm. An equation was used to calculate the scavenging rate of DPPH for each concentration of pectin solution.

The formula for calculating the scavenging rate of DPPH is as follows:(2)$$\mathrm{DPPH}\;\mathrm{radical}\;\mathrm{scavenging}\;\mathrm{activity}\left(\%\right)=\frac{A_{0}-\left(A_{c}-A_{d}\right)}{A_{0}}\times 100$$
where *A*_0_ is the absorbance of DPPH· with absolute ethanol, *A_c_* is the absorbance of the sample solution with DPPH· and absolute ethanol, and *A_d_* is the absorbance of the sample solution with absolute ethanol.

#### 2.8.2. Hydroxyl Radical (·OH) Scavenging Activity Determination

·OH radical scavenging activity was determined using the research by Hu et al. [24]. In order to initiate the reaction, the following reagents were added in sequence to the reaction system: 0.5 mL of FeSO_4_ solution (9 mM), 0.5 mL of ethanol-salicylic acid solution (9 mM), 0.5 mL of pectin sample with varying concentration gradients, 3.5 mL of distilled water, and 0.5 mL of H_2_O_2_ (8.8 mM). It is important to note that the reaction mixture was added in the specified order. The reaction mixture was thoroughly mixed and incubated at 37 °C for 15 min. When removing, shaking was avoided to prevent precipitation. The absorbance value was measured at 510 nm.

The formula for calculating the scavenging rate of ·OH radicals is as follows:(3)$$\mathrm{OH}\;\mathrm{radical}\;\mathrm{scavenging}\;\mathrm{activity}\left(\%\right)=\frac{A_{0}-\left(A_{c}-A_{d}\right)}{A_{0}}\times 100$$
where *A*_0_ is the absorbance value without pectin sample, *A_c_* is the absorbance value with different concentrations of pectin sample, and *A_d_* is the absorbance value without FeSO_4_ solution.

#### 2.8.3. Reducing Power Determination

Reducing power was determined using the method by Mzoughi et al. [25]. First of all, 0.1 mL of pectin samples with varying concentration gradients were mixed with 0.5 mL of phosphate buffer (pH 6.6) and 0.5 mL of K_3_[Fe(CN)_6_] solution (1%). Then, 0.5 mL of 10% TCA solution was added and mixed well. The mixture was left to stand at room temperature for 10 min. Next, 0.12 mL of the mixture was taken and mixed with 0.12 mL of distilled water and 25 μL of FeCl_3_ solution (0.1%). The mixture was left at 50 °C for 20 min. After 10 min, the absorbance value was measured at 700 nm.

### 2.9. Statistical Analysis

All experiments were performed in triplicate. Data processing was performed using Microsoft Excel 2016 (Microsoft Corporation, Redmond, Washington, DC, USA), and statistical analysis was conducted using SPSS 27.0 software (IBM Corporation, Armonk, NY, USA). One-way analysis of variance (ANOVA) was employed to investigate the significant differences (*p* < 0.05) among different parameters of hawthorn pectin. Results are presented as mean ± standard deviation. Graphs were plotted using Origin 2022 software (Origin Lab Corporation, Northampton, MA, USA).

## 3. Results

### 3.1. Analysis of Basic Indicators of Hawthorn Pectin

The basic indicators of hawthorn pectin prepared using different methods are shown in Table 1. The degree of esterification of all three types of hawthorn pectin is above 50%, indicating that they all belong to high-methoxy pectin. Among them, AE-HP has the highest degree of esterification, followed by WE-HP and UPAAE-HP. This indicates that acid intervention is beneficial for increasing the degree of esterification, but the involvement of ultra-high pressure has the opposite effect. The reason may be that the C-O bonds of carboxyl groups are disrupted under higher pressure in the demethylation reaction [26]. As Table 1 shows, the content of total galacturonic acid in UPAAE-HP increased, which is consistent with the research results of Zhao et al. [27]. Compared with AE-HP, it was increased by 0.71%. The reason may be that the assistance of ultra-high pressure makes it easier for galacturonic acid to leach out. However, the content of galacturonic acid in all three types of hawthorn pectin is below 65%, which does not meet the commercial pectin standards and requires further purification. The protein content in UPAAE-HP is slightly lower than in AE-HP, which may be attributed to the significant pressure causing protein denaturation, and during centrifugation, proteins might be discarded along with the precipitate. The total polyphenol content in all three types of hawthorn pectin is below 1%. The intervention of ultra-high pressure may lead to a slight decrease in polyphenol content, but the effect is not significant. In summary, the assistance of ultra-high pressure has led to a decrease in the esterification degree, protein content, and total polyphenol content of hawthorn pectin. At the same time, it has increased the total galacturonic acid content.

### 3.2. FT-IR Analysis

The FT-IR spectra of hawthorn pectin extracted using different methods are shown in Figure 1. The spectra indicate that the three types of hawthorn pectin have similar types of infrared absorption groups within the scanned range. This suggests that the intervention of ultra-high pressure does not cause significant changes in the pectin structure. However, in terms of the intensity of absorption peaks, comparing UPAAE-HP with AE-HP, the absorption peaks of UPAAE-HP are generally reduced. This suggests that ultra-high pressure alters the interactions between pectin molecules, leading to changes in the intensity of absorption peaks. This may be related to the magnitude of pressure. This is consistent with the findings of Zhang et al. [28].

All three types of hawthorn pectin exhibit a broad absorption peak in the range of 3700 to 3000 cm^−1^, which is attributed to the O-H stretching vibration of chemical groups [1]. This peak is mainly associated with the hydrogen bonding vibration modes between and within the molecules of GalA polymers [29]. The peak observed around 2932 cm^−1^ is primarily associated with the stretching vibration of C-H bonds [1]. It can be observed from the graph that the absorption peak at 2932 cm^−1^ is more prominent in AE-HP, indicating a larger vibrational amplitude for this sample. The peaks at 1745 cm^−1^ and 1627 cm^−1^ are attributed to the asymmetric stretching vibrations of the esterified carboxyl group (C-O) and the carbonyl group (C=O) in the free carboxyl group, respectively [30]. The presence of these two absorption peaks indicates that WE-HP, AE-HP, and UPAAE-HP are all pectin polysaccharides [9,31]. The degree of esterification of pectin can be calculated using the ratio of the areas of these two peaks [A_1730_/(A_1730_ + A_1630_)]. From Figure 1, it can be inferred that all three types of pectin are high-methoxyl pectin, consistent with the results obtained using titration. The absorption peak around 1442 cm^−1^ is likely due to the bending vibration of the carbon–hydrogen bond [32]. The peak observed at 1235 cm^−1^ in AE-HP indicates the presence of (-O-CH_3_) groups [33]. The absorption peaks around 1200~1000 cm^−1^ are likely due to the overlapping stretching vibrations of C-O-C and C-O-H bonds, suggesting the presence of pyranose form sugars [34]. Meanwhile, the absorption band range of 1300~800 cm^−1^ is referred to as the “fingerprint region” for pectin [4,30]. In Figure 1, slight differences in peak shapes within this wavelength range indicate slight variations in the monosaccharide composition of hawthorn pectin extracted using different methods.

### 3.3. Morphological Characteristics Analysis

The scanning electron microscope (SEM) is one of the most effective tools for observing differences in morphology and microstructure between samples [2]. Figure 2 shows the morphology of freeze-dried hawthorn pectin powders, prepared using various extraction methods, as observed at magnifications of 100× and 5.0 k× under the electron microscope. It can be observed that at low magnifications, the hawthorn pectin obtained using all three methods under optimal conditions exhibits a flake-like structure. Upon further magnification, it was observed that the surfaces were composed of interconnected spherical particles arranged in sheets.

Although there is little difference in overall morphology, there are subtle differences in their microstructures. The surface of UPAAE-HP (Figure 2C) exhibited a noticeably porous layered structure, and under high magnification (Figure 2c) enlarged spherical particles with cracks on the surface were observed, indicating the presence of voids inside, which may be related to excessive extraction pressure and rapid pressure release [35]. This is consistent with the results of Zheng et al. [11], which showed that ultra-high pressure extraction would make the surface structure of the extract more curly and porous.

On the contrary, WE-HP (Figure 2A) exhibited a dense sheet-like structure with a relatively rough surface. Under high magnification (Figure 2a), a honeycomb-like and reticular structure was observed between closely packed particles. The surface of AE-HP (Figure 2B) appeared smooth and continuous, exhibiting a regular and compact spatial structure. Compared to WE-HP, there was a slight emergence of sheet-like structures on its surface, presumably due to structural changes induced by the acid. Meanwhile, as observed in Figure 2a,b, compared to AE-HP, WE-HP exhibits a denser molecular structure, indicating that citric acid may lead to the polymerization of pectin molecules, which was consistent with the results of He [36].

### 3.4. Monosaccharide Composition Analysis

The ion chromatograms of 16 monosaccharide standards and the ion chromatograms of hawthorn pectin extracted using different extraction methods are shown in Figure 3. The composition of monosaccharides detected using ion chromatography was roughly the same in all hawthorn pectin, but their concentrations were different. It is worth noting that in the hawthorn pectin obtained through ultra-high pressure extraction, the monosaccharide composition includes Rib, and its content is 5.50 μg/mg. This indicates that the use of ultra-high pressure assistance on the basis of acid extraction will change the types and proportions of monosaccharide compositions in pectin samples. The occurrence of this phenomenon may be attributed to the further breakdown of riboflavin in the raw materials during the ultra-high pressure extraction process.

All three types of hawthorn pectin contain seven monosaccharides (Fuc, Rha, Ara, Gal, Glc, Xyl, GalA), with glucose (132.79~260.92 μg/mg) and galacturonic acid (61.88~157.30 μg/mg) being the major components of hawthorn pectin. Additionally, arabinose (26.93~56.77 μg/mg) and galactose (16.01~37.62 μg/mg) are present in lower quantities. The involvement of ultra-high pressure resulted in a slight increase in the galacturonic acid content of UPAAE-HP compared to AE-HP, but its content remains slightly lower than that of WE-HP. This is consistent with the results obtained using the carbazole-sulfuric acid method mentioned earlier. The high content of Glc in the pectin samples may be attributed to residual cellulose and hemicellulose (non-pectic polysaccharides) [3,5].

The molecular structure of pectin is mainly composed of three regions: homogalacturonan (HG), rhamnogalacturonan I (RG-I), and rhamnogalacturonan II (RG-II) [37]. The ratios of rhamnose to galacturonic acid in WE-HP, AE-HP, and UPAAE-HP are 0.10, 0.09, and 0.10, respectively. This indicates that all three types of hawthorn pectin belong to the RG-I type [38]. The content of Ara in WE-HP is relatively low compared to AE-HP and UPAAE-HP, while the content of Gal is slightly higher compared to AE-HP and UPAAE-HP. This may be attributed to the involvement of acid, which makes Ara more susceptible to degradation [39].

### 3.5. Molecular Weight Distribution Analysis

As shown in Table 2, it can be observed that different extraction methods have a significant impact on the molecular weight of hawthorn pectin. All three types of hawthorn pectin have high molecular weight (19.9–2000 kDa), which is strongly associated with the presence of a significant amount of glucose in them [36]. This also confirms the results of the pectin monosaccharide composition.

In comparison, it is evident that UPAAE-HP has the highest molecular weight. Considering the analysis of monosaccharide composition, this might be attributed to the formation of pectin–cellulose macromolecular complexes during the process of ultra-high pressure extraction, which could contribute to an increase in the molecular weight of pectin [39]. UPAAE-HP and AE-HP exhibit significant differences in molecular weight compared to WE-HP. The reason for this difference may be that under relatively acidic conditions, most high molecular weight pectins are retained. It has been reported in related literature [40] that after citric acid treatment, the molecular structure of pectin tends to polymerization, resulting in an increase in molecular weight. This is consistent with the scanning electron microscopy results.

The Mw/Mn ratio, known as the polydispersity index, reflects the width of the molecular weight distribution of polymers [30]. As shown in Table 2, the Mw/Mn ratios of the three types of hawthorn pectin are consistent, all equaling 1.02. This indicates that the molecular weight distribution of hawthorn pectin extracted using different methods is relatively narrow.

### 3.6. Antioxidant Activity Analysis In Vitro

The in vitro antioxidant activities of hawthorn pectin prepared using different methods (UPAAE, WE, and AE) were evaluated using DPPH free radical scavenging capacity, hydroxyl radical scavenging capacity, and reducing power assays.

#### 3.6.1. DPPH Radical Scavenging Activity

The scavenging effect of hawthorn pectin polysaccharides on DPPH free radicals is shown in Figure 4A. It can be observed that as the concentration of pectin solution increases, the scavenging rates of pectin solutions extracted using the three methods on DPPH radicals show an upward trend, indicating good scavenging effects. Among them, when the concentration is below 1 mg/mL, the scavenging rate of DPPH radicals by WE-HP is significantly higher than that of AE-HP and UPAAE-HP. Research [31] has shown that pectin with a high content of galacturonic acid and low molecular weight has a higher scavenging ability for DPPH radicals, which is consistent with the previous research results. At a concentration of 2 mg/mL, the scavenging effect of UPAAE-HP on DPPH radicals is comparable to that of WE-HP but higher than AE-HP. Moreover, at a concentration of 4 mg/mL, UPAAE-HP exhibits the highest DPPH scavenging rate (93.70%), approaching the scavenging effect of Vc (96.30%), which is in agreement with the results of Zheng et al. [11] and Joo et al. [41], whose studies also showed that ultra-high pressure assisted extraction could effectively enhance the antioxidant effect.

#### 3.6.2. Hydroxyl Radical Scavenging Activity

The scavenging effects of hawthorn pectin polysaccharides on hydroxyl radicals are depicted in Figure 4B. It can be observed that with the increase in the concentration of pectin solution, the scavenging rates of hydroxyl radicals by pectin solutions extracted through the three methods show an upward trend. In particular, at concentrations greater than 2 mg/mL, the scavenging capacity of hydroxyl radicals by UPAAE-HP is second only to AE-HP. This may be related to the extracted content of galacturonic acid. At the same concentration, samples with higher content of galacturonic acid are prone to forming clusters in the solution. The presence of this state reduces the contact surface between pectin and hydroxyl radicals, leading to a decrease in the scavenging capacity for hydroxyl radicals [23].

#### 3.6.3. Reducing Power

The reducing power of hawthorn pectin polysaccharides is shown in Figure 4C. It can be observed that the absorbance increases with the concentration of hawthorn pectin solution, indicating a dose-dependent relationship between the reducing power of WE-HP, AE-HP, and UPAAE-HP and their concentrations. When comparing the three kinds of hawthorn pectin, there was little difference in the reducing ability, but there was a significant difference compared with the ascorbic acid (V_C_) control group, indicating that the reducing ability of hawthorn pectin was relatively low.

## 4. Conclusions

In this study, three pectin polysaccharides (UPAAE-HP, WE-HP, and AE-HP) were successfully extracted from the hawthorn powder and compared. To delve into the differences among the extraction methods, a comprehensive analysis was conducted, covering basic indicators, FT-IR spectrum, SEM image, monosaccharide composition, molecular weight, and in vitro antioxidant activity.

The results of basic Indicators indicate that, with the assistance of ultra-high pressure, the DE, protein content, and TPC of hawthorn pectin are reduced to varying degrees. However, the GalA content increases. The FT-IR spectrum indicates that there is no significant difference in the assistance of ultra-high pressure compared to the other two methods, but there is a slight reduction in the intensity of absorption peaks. SEM analysis reveals that all three types of hawthorn pectin exhibit a sheet-like structure. However, UPAAE-HP displays a noticeably porous layered structure on its surface. The analysis of monosaccharide composition reveals that all three types of hawthorn pectin contain Fuc, Rha, Ara, Gal, Glc, Xyl, and GalA. They all belong to the RG-I type pectin. The molar ratio of GalA to Ara increased in UPAAE-HP, and the ultra-high-pressure technology in this process also led to the release of Rib from the raw materials. UPAAE-HP has the highest molecular weight, reaching 558.45 kDa. In terms of antioxidant activity, UPAAE-HP exhibits strong antioxidant activity, and it shows a dose-dependent relationship, especially in the clearance of DPPH free radicals.

All in all, compared with hawthorn pectin prepared using other extraction methods, UPAAE-HP has the advantages of lower esterification degree, total polyphenols, and high antioxidant activity. Additionally, it can also thoroughly release the monosaccharides in hawthorn powder and destroy the structure more thoroughly. Therefore, ultra-high pressure assisted is a green and effective extraction method for hawthorn pectin. In order to have a better understanding of the processing characteristics and other functional characteristics of hawthorn pectin, further in-depth research is needed in the future.

## Figures and Tables

**Figure 1 foods-13-00983-f001:**
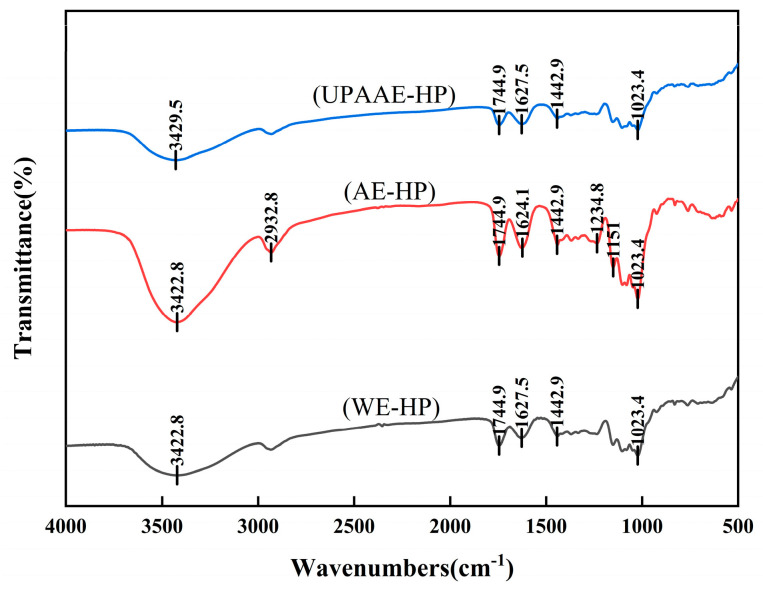
FT-IR spectra of hawthorn pectin extracted using different methods.

**Figure 2 foods-13-00983-f002:**
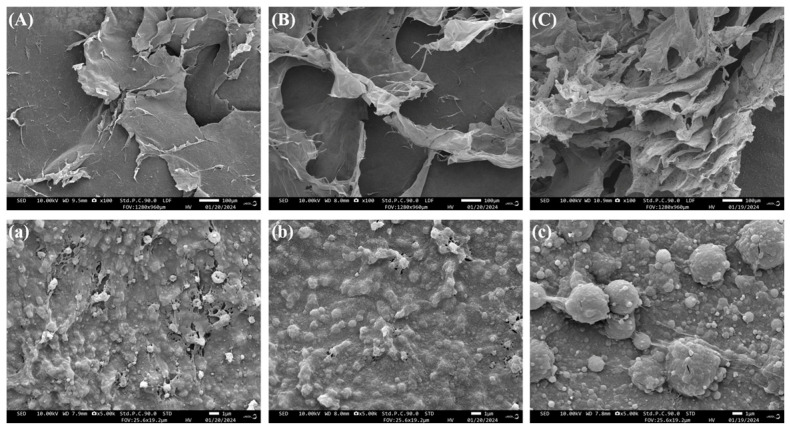
SEM images of WE-HP (**A**,**a**), AE-HP (**B**,**b**), and UPAAE-HP (**C**,**c**); (**A**–**C**) 100× magnification; (**a**–**c**) 5.0k× magnification.

**Figure 3 foods-13-00983-f003:**
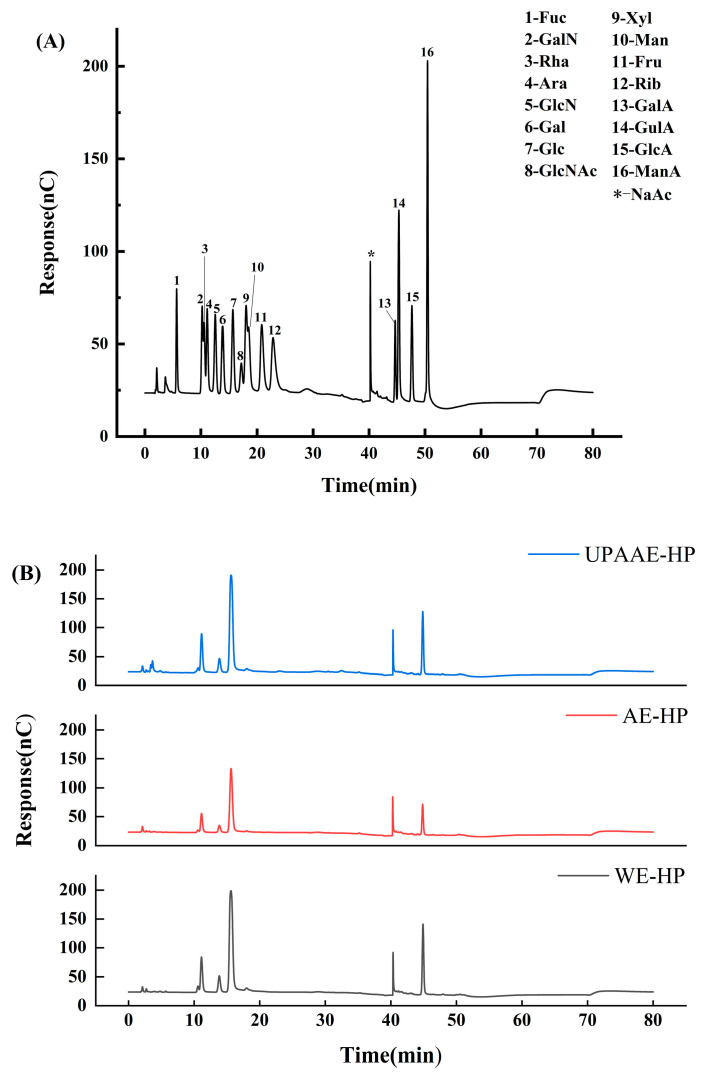
(**A**) Ion chromatogram of monosaccharide mixed standard; (**B**) ion chromatogram of hawthorn pectin extracted using different methods.

**Figure 4 foods-13-00983-f004:**
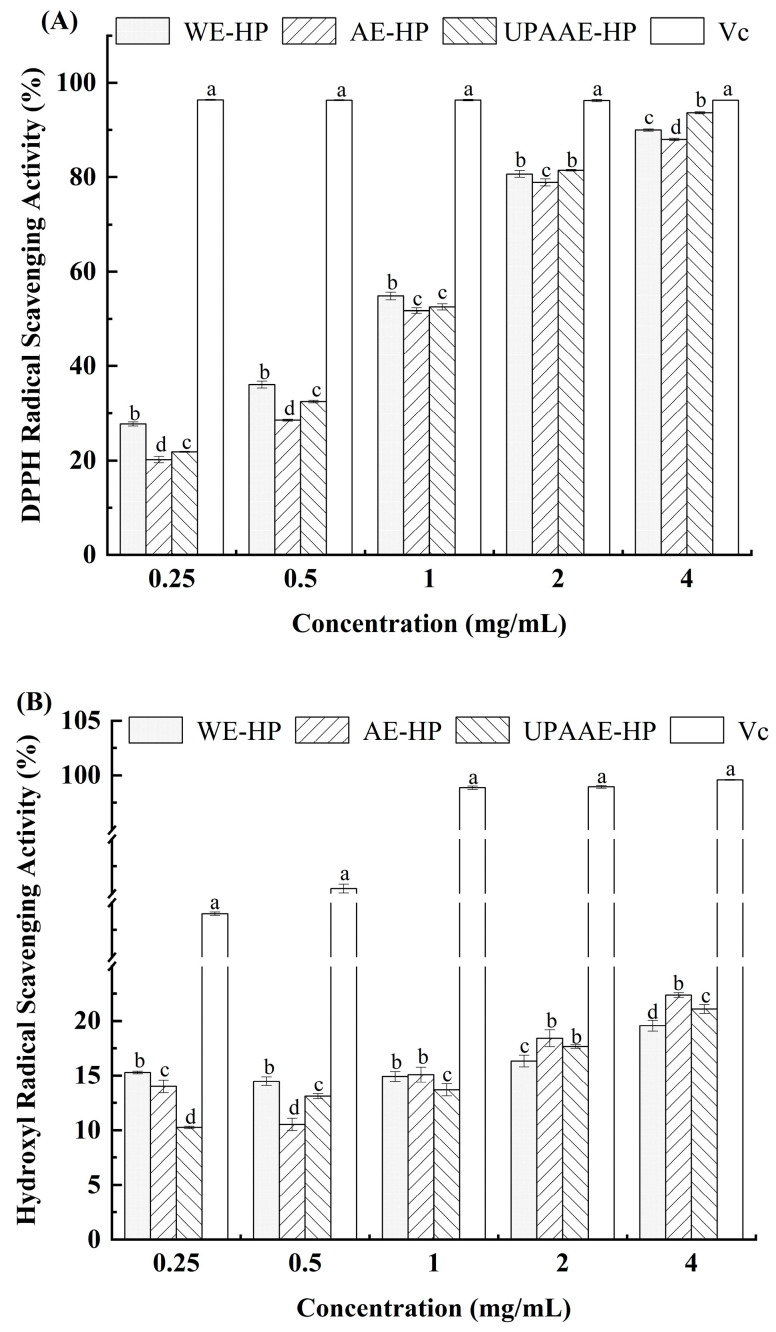
(**A**) Scavenging activity of hawthorn pectin polysaccharides on DPPH free radicals; (**B**) scavenging effect of hawthorn pectin polysaccharides on hydroxyl free radicals; (**C**) total reducing power of hawthorn pectin polysaccharides. Note: different lowercase letters on the shoulders of the same indicator indicate significant differences (*p* < 0.05).

**Table 1 foods-13-00983-t001:** Basic indices of hawthorn pectin extracted using different methods.

Basic Indices	WE-HP	AE-HP	UPAAE-HP
DE/%	56.90 ± 2.44 a	58.42 ± 3.97 a	55.84 ± 2.12 a
GalA/%	55.83 ± 0.90 a	52.57 ± 0.74 b	53.28 ± 0.44 b
Protein content/%	5.30 ± 0.84 c	10.50 ± 0.64 a	8.71 ± 0.72 b
TPC/%	0.96 ± 0.03 a	0.98 ± 0.04 a	0.92 ± 0.02 b

Note: Different lowercase letters in the same line indicate statistically significant differences (*p* < 0.05).

**Table 2 foods-13-00983-t002:** Molecular weight of hawthorn pectin extracted using different methods.

Samples	Mw/(KDa)	Mn/(KDa)	Mw/Mn
WE-HP	382.50	373.80	1.02
AE-HP	508.37	496.79	1.02
UPAAE-HP	558.45	545.74	1.02

## Data Availability

The original contributions presented in the study are included in the article, further inquiries can be directed to the corresponding author.

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
