# Peer review of "The Physicochemical Properties and Antioxidant Activities of the Hawthorn Pectin Extracted Using Ultra-High Pressure Assisted Acid Extraction (UPAAE)"

_foods, 2024, doi:10.3390/foods13070983_

Round 1
Reviewer 1 Report
Comments and Suggestions for Authors
The paper is interesting and the results presented are important for the scientific community, but I have some concerns about the paper which need to be addressed. Please see comments below.
Abstract contains abbreviations which were not explained at first mention. Please add full name before mentioning the abbreviation for the first time.
P1, L37: why is CHEN capitalized? Please correct.
P2, L54: Which limitations? Please list them.
P2, L55 - 56: Please rephrase the sentence T"he initial design of ultra-high pressure technology was primarily for sterilization in the food".
Introduction: In P2, L45, you say mention the fact that pectin production falls short of the demand, and new sources of pectin are used. Later on, you mention the drawbacks of current extraction technologies and list the benefits of UPAAE. My question is, since the UPAAE has to be done at high pressures using acid reagents at low pH, where is the feasibility of this method? It seems to me that the high costs are something that hinders its application in the industry and therefore, its production on a large industrial scale. Please elaborate the drawbacks and the possibilities of UPAAE use on a larger scale in the Introduction section.
P2, L70: please list the latin name and describe the hawthorn in more detail (year and place of harvest, was it obtained fresh or dried, initial moisture content etc.).
P2, L70 - 76: Please list the purity, producers, city and the state of origin of all used chemical and reagents.
P2, L77 - 93, P3, L 100 - 104, L108 - 114; L 121 - 130, P4, L165-169; P4, L 184 - 190; P5, L208 - 214; P5, L223 -229; P5, L238 - 243: The methods should not be described like cookbook instructions, but should be written in passive. Please correct.
P3, L 140 - 141; P4, L148 and159: There is no need to put the trendline equation and fit in the manuscript. Please remove it.
P3, L203 - 205: You state that you measured in vitro antioxidant activity. What does this include? In vitro digestion or what? Usually, the in vitro digestion simulation is done, but that has to include different digestion steps. Or maybe, different pH levels which correspond to those of the digestive system. In your case, you did not elaborate any of it in the materials and Methods section, so I do not understand what was actually done and what is meant by "in vitro" in your case. Please elaborate.
P6, L244 - 247: What type of statistical analysis? Which test was done? At which probability level? Since you had a low number of samples, was the data previously tested for normality and by which method in order to ensure the appropriate statistical test was used? Please elaborate.
Table 1 has no statistical significance data in it.
Comments on the Quality of English Language
The English quality definitely needs to be approved, especially the style of describing the analysis methods.
Reviewer 2 Report
Comments and Suggestions for Authors
The comment for authors:
1. The authors should check text for typos and grammatical errors.
2. Overall conclusion is missing in the Abstract section.
3. The authors should specify the disadvantages of the techniques: “The properties of pectin polysaccharides obtained by ultrasound-assisted and microwave-assisted methods are uneven, leading to certain limitations in the pectin extraction process.”
4. The applied extraction technique needs to be more explained in the introduction section.
5. Moisture content of dried sample material need to be provided.
6. The obtained results should be more discussed in terms of comparison with previously published data on other pectin sources. What is the advantage/significance of pectin isolated from hawthorn in comparison to other sources?
7. Statistical analysis should be included in Table 1 and 2.
8. Practical application and future perspective have to be more highlighted in the Conclusion section.
